

# Brief communication: Not as dirty as they look, flawed airborne and satellite snow spectra

Edward H. Bair[1], Dar A. Roberts[2], David R. Thompson[3], Philip G. Brodrick[3], Brenton A. Wilder[4], Niklas Bohn[3], Christopher J. Crawford[5], Nimrod Carmon[3], Carrie M. Vuyovich[6], and Jeff Dozier[2]

[1]Leidos, Reston, VA, USA
[2]University of California, Santa Barbara, CA, USA
[3]Jet Propulsion Laboratory, California Institute of Technology, Pasadena, CA, USA
[4]Boise State University, ID, USA
[5]US Geological Survey, Earth Resources Observation and Science Center, SD, USA
[6]National Aeronautics and Space Administration, MD, USA

*Correspondence to*: Edward Bair (baire@leidos.com)

**Abstract**

Key to the success of spaceborne missions is understanding snowmelt in our warming climate, having implications for nearly 2 billion people. An obstacle is that surface reflectance products over snow show an erroneous hook that often shows sharp
decreases in the visible wavelengths. This hook is sometimes mistaken for soot or dust but can result from three artifacts: 1) a background reflectance that is too dark; 2) an assumption of level terrain; 3) or differences in optical constants of ice. Sensor calibration and directional effects may also contribute. Solutions currently being implemented address these artifacts.

## 1. Introduction

Current and future hyperspectral missions such as the Earth Surface Mineral Dust Source Investigation (EMIT), Precursore
Iperspettrale della Missione Applicativa (PRISMA), or Surface Biology and Geology (SBG), offer improved spectral resolution and fidelity, yet surface reflectance products lag sensor advances. Of the terms in the energy balance, snowmelt is most sensitive to albedo. Because of snow's importance as a water resource, it is among the "Most Important" objectives for future NASA missions, requiring measurement and modeling accurately enough to close the surface radiation balance to within 10% of the absorption (National Academies of Science, Engineering and Medicine, 2018). The prevalent erroneous *hook*,
where in the decreasing case, the surface reflectance sharply decreases with decreasing visible wavelength, e.g., brighter at 600 nm than at 400 nm, compromises this "Most Important" objective. The decreasing hook is easily mistaken for the presence of light absorbing particles (LAPs) such as soot or dust. This Brief Communication shows examples of the hook, analyzes the causes, and offers solutions that are being implemented. The objective is to document the cause of these common hooking errors so they can be prevented, thereby allowing scientific goals to be met for current and future missions.



## 2. Examples of erroneous hooking

Standard surface reflectance products are rife with hooking errors. Figure 1a shows modeled spectra for clean snow. Figure 1b shows modeled spectra for dirty snow, which include a legitimate hook in the visible wavelengths. Figure 1c shows the problematic hook in a surface reflectance retrieval from PRISMA compared to an in situ spectrometer measurement.

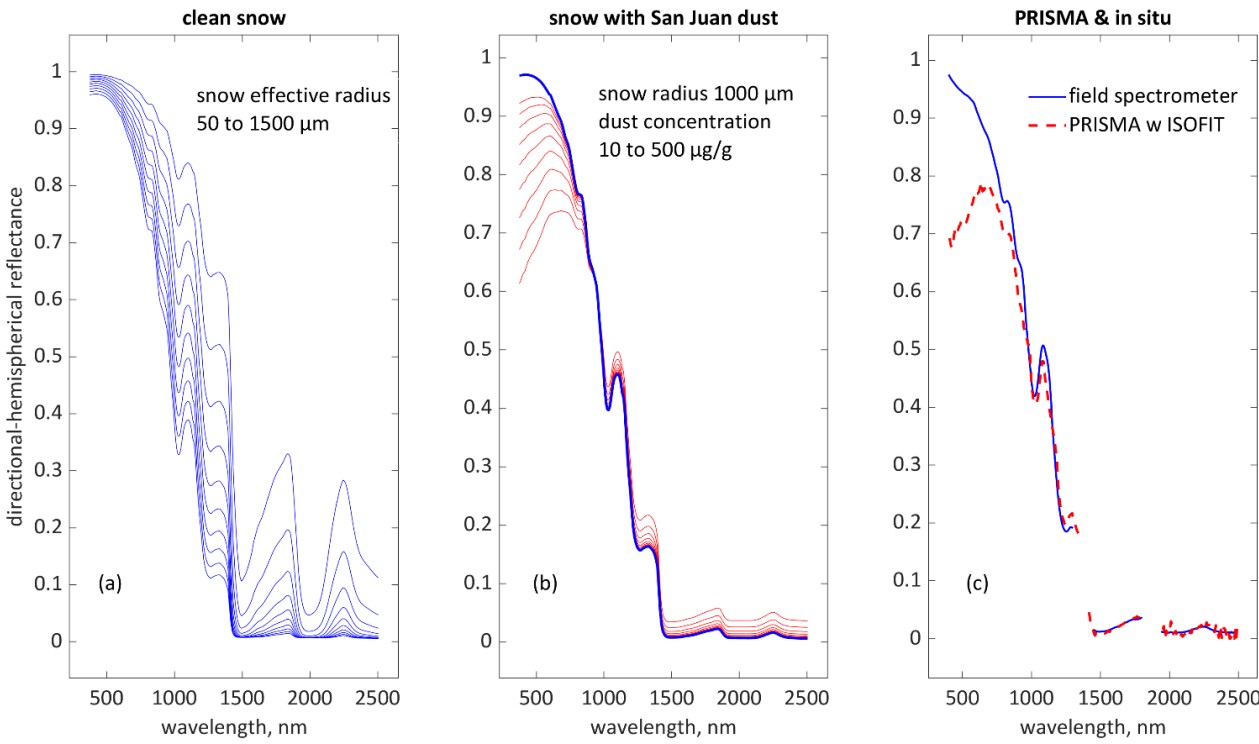

*Figure 1: (a) Spectra for clean snow modeled with SNICAR-AD V4 (Whicker et al., 2022); (b) modeled spectra for snow with San Juan dust* (Skiles et al., 2017) *of radii 1.25-2.5 µm; (c) field spectrometer measurements of snow compared to PRISMA surface reflectance (Townsend et al., 2023).*

Surface reflectance products show suspicious hooking in AVIRIS-NG (Green et al., 2023), PRISMA (Townsend et al., 2023), EMIT (Green, 2022), and Collection 2 Landsat 8/9 (Crawford et al., 2023). Often, the hook can be diagnosed visually, without modeling. For example, fine-grained but dirty snow is suspicious. This improbable, but commonly seen combination in surface reflectance products, shows as hooking in the visible spectrum combined with indicators of fine-grained snow in wavelengths beyond 1000 nm. Likewise, a peak close to 1.0 in any wavelength in the presence of dust or soot is unlikely.

## 3. Approach

The bihemispherical spectral reflectance of snow, commonly called spectral albedo, at a wavelength $R_\lambda$ is:



$$R_\lambda = \frac{D_\lambda}{I_\lambda} \tag{1}$$

where $D_\lambda$ is the reflected radiation and $I_\lambda$ is the combined direct and diffuse irradiance. Non-Lambertian behavior of snow has been known for over 70 years (Middleton and Mungall, 1952), however bidirectional reflectance distribution models struggle over rough surfaces, such as ablation hollows, as the viewing geometry causes shadowing (Bair et al., 2022). Thus, because of the unknown lighting geometry over rough surfaces, albedo is used here to model the hooking. This albedo can be adjusted for atmospheric effects to estimate the reflectance at Earth's surface, the adjustment involving approximations for both the

numerator and denominator in Eq. (1). Instead, a simpler approach is taken where the denominator can be re-written, omitting the $\lambda$ for readability,

$$I(\delta, \mu_s, r_b) = I_{direct} + I_{diffuse} \tag{2}$$

where $I_{direct}$ and $I_{diffuse}$ are the direct and diffuse irradiance that depend on atmospheric properties $\delta$ which include aerosol, water vapor concentration, optical thickness, target altitude, air temperature, terrain configuration, and many others. Additionally, $I_{diffuse}$ depends on the spectral reflectance $r_b$ of the areas adjacent to the target, caused by atmospheric

scattering of reflected radiation. The numerator in Eq. (1) contains all the terms of the denominator, but also terms for the target direct and diffuse reflectance, $R_{direct}$ and $R_{diffuse}$,

$$D(\gamma, \mu_s, \delta, r_b) = R_{direct} \times I_{direct} + R_{diffuse} \times I_{diffuse} \tag{3}$$

with snow properties $\gamma$ (grain radius and light absorbing particle concentration).

**Hook caused by atmospheric correction algorithm**

Widely used atmospheric radiative transfer codes—e.g., MODTRAN, 6S, SMARTS, libRadtran—allow for a variety of

background reflectance $r_b$ options, from constant values to user-defined spectra to spectral libraries, or even spectral mixtures. Concentrating on the background reflectance, the spectral reflectance in a snow-covered region can be modeled as

$$R = \frac{D(r_{b,snow}|\gamma, \mu_s, \delta)}{I(r_{b,snow}|\mu_s, \delta)} \tag{4}$$

where the numerator is calculated with Eq. (3), the denominator is the sum of the direct and diffuse irradiances, and $r_{b,snow}$ is the spectral background reflectance in the area around the snow-covered pixel of interest. Parameterizations differ widely, but for operational products, instead of using the $r_{b,snow}$ spectra, which varies with wavelength (Figure 1), a constant $r_b$ value

similar to Earth's planetary albedo, 0.25 - 0.30, is typically used. The decreasing hook error can be simulated by recognizing that the background reflectance $r_b$ is too dark in snow-covered terrain. To model the decreasing hook with Eq. (4), an $r_{b,dark} =$



0.25 is used in the numerator, while $r_{b,snow}$ is used in the denominator, signifying that the downwelling radiation is correctly modeled, but the upwelling radiation is incorrectly modeled,

$$R_{upwelling\ error} = \frac{D(r_{b,dark}|\gamma, \mu_s, \delta)}{I(r_{b,snow}|\mu_s, \delta)} \tag{5}$$

To simulate an increasing hook, the error is inverted, with the downwelling radiation incorrectly modeled, but the upwelling radiation correctly modeled,

$$R_{downwelling\ error} = \frac{D(r_{b,snow}|\gamma, \mu_s, \delta)}{I(r_{b,dark}|\mu_s, \delta)} \tag{6}$$

**Hook caused by assuming flat topography**

Likewise, because diffuse irradiance is weighted toward the blue end of the solar spectrum, errors in the modeled spectral shape will occur when topography is assumed flat. This type of error can be modeled assuming a slope that faces either toward or away from the Sun, where the numerator in Eq. (1) is modeled correctly as the slope angle changes, but the denominator uses direct irradiance for a level surface. With $\mu_S$ the cosine of the illumination angle on a slope and $\mu_0$ the illumination cosine on a level surface, the apparent terrain reflectance is

$$R_{terrain} = \frac{R_{direct}(\gamma, \mu_s) \times I_{direct}(\delta, \mu_s) + R_{diffuse}(\gamma) \times I_{diffuse}(\delta, r_b)}{I_{direct}(\mu_0, r_b) + I_{diffuse}(\delta, r_b)} \tag{7}$$

For modest slopes, less than about 30°, facing open terrain, the terrain view factor can be ignored (Dozier, 2022, Eq. 2).

**Hook caused by refractive index of ice in short wavelengths**

A third, minor cause of the hooking depends on values of the imaginary part of the complex refractive index of ice, i.e., the absorption coefficient. Ice is exceptionally transparent in the wavelengths below 500 nm range and there is disagreement in the literature of its optical properties in this range (Warren, 1984; Warren and Brandt, 2008; Picard et al., 2016). In any case, the hooking effects due to variations in the absorption coefficient are small compared to the atmospheric and terrain correction errors.

**4.  Discussion and conclusion**

Figure 2 summarizes hooking causes, which were modeled using SNICAR-AD-V4, SMARTS version 2.9.9, and Eqs. (4) - (7). The same results come from Mie theory and two-stream radiative transfer (Bair et al., 2021) instead of SNICAR.



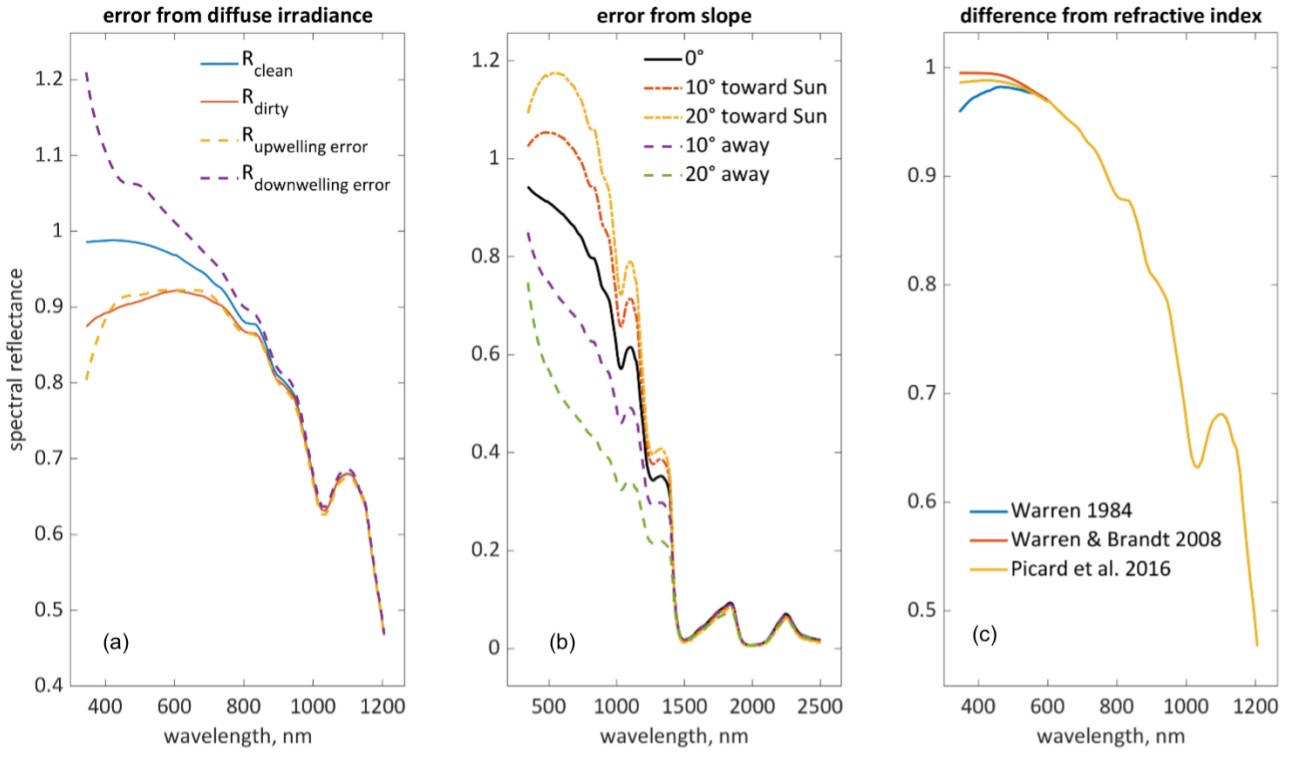

*Figure 2: Hooking in the modeled albedo of 200 μm snow, $\mu_0 = 0.6$. (a) Upwelling and downwelling atmospheric correction errors; upwelling and downwelling errors are for clean snow; dirty snow includes 100 ppmw of San Juan dust of radius 1.25-2.5 μm. (b) Spectral shape changes due to lack of terrain correction, with downwelling direct/diffuse radiation errors when not adjusted for slope angles 0° (no error) to 20°. (c) Differences due to optical properties of ice.*

In summary, the hooking in clean and fully-covered snow pixels is caused by: 1) Assumed background reflectance that is too dark; 2) lack of terrain correction; 3) differences in optical constants. Picard et al. (2020) have documented previously the errors in measuring snow reflectance over sloping terrain, but the other two causes of hooking in the spectra have not previously been documented. Two additional causes of hooking that are suspected, but not confirmed through modeling, are sensor calibration and directional effects. For sensor calibration, the blue wavelength range is often challenging to calibrate, because laboratory sources are much dimmer in those wavelengths relative to the solar profile (Helmlinger et al., 2016). Any out of band response will result in excessive blue signal during calibration, causing an inaccurate estimate of calibration coefficients, and a resulting overestimate of instrument sensitivity. Snow, because of its brightness, often lies near the upper end of airborne and spaceborne spectrometers' dynamic range, making it susceptible to saturation and associated nonlinear effects. This error, which could cause increasing or decreasing hooking, is particularly difficult to model given often unpublished calibration data. Directional effects for angular new snow may cause an increasing hook, seen in measured spectra (e.g., Painter and Dozier, 2004), especially in the forward direction (away from the sun), towards the limb (high viewing zenith angle), and when the



sun is low in the sky (high solar zenith angle). However, in the region of optimal remote sensing, i.e., low solar zenith and viewing angles, hooking effects from anisotropic snow reflectance are minimal.

To address the three modeled causes, the following are recommended: 1) use an atmospheric correction with an appropriate background reflectance; 2) correct for terrain illumination angle, but be aware of error propagation in slope and aspect (Dozier et al., 2022); and 3) use updated optical constants for ice (Picard et al., 2016) when performing inversions to solve for snow covered-area, grain size, and LAP concentration. For standard surface reflectance products, 1 & 2 need to be addressed in processing workflows, or perhaps through on-demand products. For example, in the EMIT processing chain based on the ISOFIT software package (Thompson et al., 2024) appropriate background assumptions are used, and terrain-corrected reflectances are now supported (Carmon et al., 2022).

**Code and data availability**

The SNICAR-AD-V4 (Whicker et al., 2022) and SPIReS (Bair et al., 2021) codes are available in public repositories identified in those articles.

**Supplement**

None

**Author contributions**

Author contributions are according to CRediT taxonomy. EHB performed all 14 contributor roles. JD contributed to writing (original draft, review & editing) and software (code for analyzing topography). DAR contributed to investigation and methodology. DRT contributed to investigation, methodology, and writing (original draft, review & editing). PGB contributed to investigation and methodology. BAW contributed to investigation (field measurements for Figure 1) and methodology. NB contributed to investigation and writing (original draft, review & editing). CJC contributed to conceptualization, methodology, and editing. NC contributed to investigation and methodology. CMV contributed to writing (review and editing) and project administration.

**Competing interests**

EHB and CMV are members of the Editorial Board at *The Cryosphere*.

**Acknowledgements**

Any use of trade, firm, or product names is for descriptive purposes only and does not imply endorsement by the U.S. Government.



**Financial support**

This research was supported by the Broad Agency Announcement Program and the Cold Regions Research and Engineering Laboratory (ERDC-CRREL) under contract no. W913E523C0002 and the National Aeronautics and Space Administration (grant nos. 80NSSC20K1722, 80NSSC20K1349, 80NSSC21K0620, 80NSSC24K0824). Brenton Wilder was supported by the NASA FINESST program. This research was also supported by the U.S. Geological Survey National Land Imaging Program's Sustainable Land Imaging Phase 2 imaging spectroscopy research and development project (USGS BASIS+:
GX22ED00TYG).

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
