# Peer review of "Brief communication: Not as dirty as they look, flawed airborne and satellite snow spectra"

_EGUsphere, 2024_

## Author Response (AR1)

Response to Reviewer 1

Referee comments are in blue, authors' original responses are in red, and additional responses are in dark red.

The manuscript by Bair et al. is focused on the erroneous representation of snow reflectance spectra in airborne and satellite data. While I find this topic very interesting and current, the content of the manuscript is quite redundant with another manuscript currently under evaluation in this same journal (https://egusphere.copernicus.org/preprints/2024/egusphere-2024-1020/) from the same group. I have already reviewed that manuscript, so I address the authors to my comments (RC1).

In general, I would highly suggest to merge these two submissions during the revision of Bohn et al. 2024. I leave to the Editor the decision on this point. If the authors prefer to keep this manuscript as a separate submission, a major review is needed before publication.

We thank Referee #1 for their careful review. We are aware of the issue with redundant information in both manuscripts, which sometimes happens while manuscripts are under simultaneous review. We have discussed this issue with the co-authors of both manuscripts and the Handling Editor for this manuscript and will remove the overlapping section and verbiage from Bohn et al. (2024) during its revision. These will be replaced with a citation to this discussion paper.

See revised Bohn et al. (2024) manuscript accepted for publication and citation to it on l 108.

Hereafter, I will detail my major concerns.
The authors build their argument on one (n=1) spectrum from PRISMA showed in Figure 1.

That is not correct. The hooking is observable across many surface reflectance products.

From lines 38-39 "Surface reflectance products show suspicious hooking in AVIRIS-NG (Green et al., 2023), PRISMA (Townsend et al., 2023), EMIT (Green, 2022), and Collection 2 Landsat 8/9 (Crawford et al., 2023)."

The Brief Communication format limits the number of figures to 3. We already have 2 multi-part figures, so we will add one additional multi-part figure showing erroneous hooking in clean snow on flat terrain. We will show erroneous hooking in: (a) AVIRIS-NG; (b) EMIT; and (c) Landsat 8/9.

See new Figure 2a-c which shows the erroneous hooking for EMIT (a), AVIRIS-NG (b), and Landsat 8 (c).

Furthermore, this spectrum is derived from a atmospheric-topographic correction that itself can introduce erroneous hook in snow reflectance.

Yes, that is the point of this manuscript.

At least, I would ask the authors to provide a comparison with standard L2(C-D) products from PRISMA.

The hooking is present there as well. See our response below to the question about which PRISMA processor was used for Figure 1.

A recent paper (Di Mauro et al. 2024) provided an evaluation of PRISMA reflectance and radiance products for different snow conditions. Same holds for Ravasio et al. (2024). In that cases, no clear hook is displayed in snow reflectance spectra. Which PRISMA processor has been used for generating the plot in Figure 1? When data have been downloaded from the ASI portal? In fact, several improvements have been made in the latest PRISMA processor (v_4_1_0_02_05). For example, Kokhanovsky et al. (2022) is based on an earlier version of the processor, and a downward hook is sometimes displayed in an area with expected clean snow (i.e. upper portion of the Nansen Ice Shelf, Antarctica).

The data are PRISMA L1 TOA, downloaded September 2022. In Kokhanovsky et al. (2022), the PRISMA L2D "processor" was used. Instead, we used the SISTER terrain-naïve "processor", so it should not depend on any updates to PRISMA processing scheme.

If the authors want to show that the hook is widespread, they should provide more evidence (e.g. different snow types, different latitudes, different sensors, etc.).

Agree, we will show the hooking for other surface reflectance products above as an additional figure.

See new Figure 2a-c.

Furthermore, they should provide evidence that the snow was clean (low concentration of impurities) at the ground.

For the Idaho site in Figure 1, snow spectra were collected at this same site on 2 Feb, 9 Feb, 23 Feb, and 10 Mar. There was no signal of LAP, which is clear from Figure 1.

Here are details for the figure we will add.

For AVIRIS-NG, we have an example from 19 Mar 2021 over Grand Mesa CO. We do not have in situ spectroscopic measurements, but they aren't needed. Instead, a snowpit was dug the day before the flight as part of the SNOWEX campaign and we have careful pit measurements and snowpack surface photos showing a clean snowpack with no evidence of dust or other impurities, which would have been noted in the pit. In contrast, the AVIRIS-NG spectra from that date show dramatic hooking indicative of several hundred ppm dust.

Response to Reviewer 1

For EMIT, we have a scene near Mammoth Mountain from 20 Feb 2023 on a flat lake showing the hooking and a terrain-corrected broadband albedo measurement of 0.80 at CUES indicating clean new snow, as detectable dust-covered snow for this region at this time would be highly unusual.

For Landsat 8, we have co-located and co-incident in situ spectroscopic measurements from a flat lake on Mammoth Mountain from 8 Apr 2021 showing erroneous hooking.

See new Figure 2a-c & added text, l 40-41 about how observations are from level and open areas with 100% fSCA and an optically thick snowpack. Figure 2a,b show a strong hook that cannot be fit with modeled LAPs. Nearby snow observations do not show visible albedo degradation, meaning that the snow may have contained small concentrations of LAPs–even snow in remote parts of Antarctica has detectable soot (Warren and Clarke, 1990)–but at levels that would not cause the dramatic hook seen in Figure 2a,b. Figure 2c is comparable to Figure 1c in that measurements from a field spectrometer do not show the hooking in the surface reflectance product.

Further information on the properties of snow at the surface is needed. I see that they reference to Townsend et al. (2023) dataset, and I learnt about the SISTER initiative. This should be described in detail in this manuscript as well. How many pixels have been averaged?

4 pixels over Idaho City Football Field. Location can be found here: 43.8387, -115.8293

Was snow flat in that area?

1.3 - 2.6 degrees. Yes.

During which period field data have been collected?

Most of the winter. But for this one you are showing, PRISMA is Feb 10 and ASD is Feb 9.

Which spectrometer and protocol have been used for field spectroscopy measurements?

-  ASD FieldSpec4
- Held bubble level 1 m above the surface
- We used the bare fiber with no fore-optic attachment for this measurement.
-  +/- 1 hour around solar noon
-  Measurements for Feb 9 were completely cloud free.

We are happy to provide this information to the Reviewer, and as these reviews are open will be accessible to any interested person, but will omit from the manuscript given it is a Brief Communication.

**Response to Reviewer 1**

In the title, I read that the manuscript is about satellite and airborne sensor. Throughout the manuscript those airborne sensors are not detailed. Can you provide evidence of hooking from airborne sensors (e.g. AVIRIS, APEX etc.)?

Yes, we will provide evidence of hooking from AVIRIS-NG, as mentioned above

See Figure 2B.

Further still on the title: If the hook is located below 500nm, likely snow will not "look" dirtier, at least from a correct RGB representation.

Unclear on this comment. Blue light < 500 nm is where most of the issue is. If the snow in polluted with LAPs, it will appear dirtier to the naked eye.

Note the hooking is pronounced throughout the visible spectrum in all of the examples (Figure 1c, Figure 2a-c).

In line 31, I read: "Standard surface reflectance products are rife with hooking errors", but no references either evidence of this hooking errors is detailed. I strongly encourage the authors to go more in detail on this error. Please, see my comments to Bohn et al. 2024 on this topic.

See above

Figure 1c & Figure 2a-c now show erroneous hooking in 4 aerial and spaceborne sensor surface reflectance products.

Line 94-95: these conclusions strongly overlaps with Section 5.1 ("the blue hook") in Bohn et al. 2024.

See above

Bohn et al. (2024) now reference this manuscript.

References:

Bohn, Niklas and Bair, Edward H. and Brodrick, Philip G. and Carmon, Nimrod and Green, Robert O. and Painter, Thomas H. and Thompson, David R., The Pitfalls of Ignoring Topography in Snow Retrievals: A Case Study with Emit. Available at SSRN: https://ssrn.com/abstract=4671920 or http://dx.doi.org/10.2139/ssrn.4671920

Response to Reviewer 1

Di Mauro, B., Cogliati, S., Bohn, N., Traversa, G., Garzonio, R., Tagliabue, G., et al. (2024). Evaluation of PRISMA products over snow in the Alps and Antarctica. Earth and Space Science, 11, e2023EA003482. https://doi.org/10.1029/2023EA003482

Kokhanovsky A, Di Mauro B and Colombo R (2022) Snow surface properties derived from PRISMA satellite data over the Nansen Ice Shelf (East Antarctica). Front. Environ. Sci. 10:904585. doi: 10.3389/fenvs.2022.904585

Ravasio, C., Garzonio, R., Di Mauro, B., Matta, E., Giardino, C., Pepe, M., et al. (2024). Retrieval of snow liquid water content from radiative transfer model, field data and PRISMA satellite data. Remote Sensing of Environment, 311, 114268. https://doi.org/https://doi.org/10.1016/j.rse.2024.114268

Warren, S. G. and Clarke, A. D.: Soot in the atmosphere and snow surface of Antarctica, Journal of Geophysical Research: Atmospheres, 95, 1811-1816, 10.1029/JD095iD02p01811, 1990.

Referee comments are in blue, authors' original responses are in red, and additional responses are in dark red.

This brief communication documents the root cause of an erroneous "hook" observed in the visible wavelengths of measured snow reflectance spectra from airborne and satellite imaging spectrometers, which is often mistaken for dirty snow. This phenomenon has been documented in recent papers and is something I have observed and documented in aerial Airborne Coastal Observatory data collected over rugged terrain using ATCOR4 atmospheric/topographic correction (Donahue et al., 2023). To my knowledge, this is the first paper that specifically investigates the cause of this issue for aerial and satellite platforms and breaks it down into multiple possible components. Specifically, I find the results shown in Figure 2 to be a valuable contribution and visualization for the community. Given the numerous current and forthcoming spaceborne imaging spectrometer missions, this is a timely communication that will help raise awareness of and provide solutions for this commonly observed artifact. The communication is well-written, and the modeling methods are sound.

We thank Chris Donahue for this review and for reiterating the prevalence of this problem. We would add a reference to Donahue et al. (2023), but we are already at the 20 reference limit.

I recommend publication following consideration of the following comments.
1. In cases where hooking is caused by the atmospheric correction algorithm, a few more details are needed to describe how the background snow reflectance spectra is used to correct downwelling and upwelling radiation. It is noted that the background snow spectrum is spectrally varying, while the dark reflectance is spectrally constant. Given this, I would expect to see differences in Figure 2 beyond 900 nm for the two error cases (dashed lines) when compared to the two unflawed spectra (solid lines), but the spectra appear to overlap each other.

It's a good point. We will test, but we expect little difference to the path radiance term from changes to the background snow grain size. This is because wavelengths > 900 nm are less sensitive to atmospheric scattering.

Confirmed. The differences are negligible.

I would also expect possible differences into the SWIR region which is not shown in the figure. Does the commonly used constant background reflectance cause artifacts in other regions of the spectra that could be a concern? Also, how does one select an appropriate background spectrum?

See above

2. NASA's goal, as stated in the introduction, is to accurately measure/model absorption within 10%. How much error could this hooking artifact introduce to a broadband albedo measurement? A brief quantitative assessment of this impact would increase the impact of the brief communication. Since solar irradiance is lower in the 350-450 nm range—where the hook is steepest—the resulting broadband albedo error, when convolved with spectral irradiance, would be smaller compared to error in longer visible bands.

Good point. Total error will depend on each of the three factors and their magnitude (e.g., slope aspect and angle), but we can estimate what we think the average and maximum errors would be.

See added text in abstract l 15 "...causing up to 33% per band and 11% broadband reflectance errors" and l 43-44

3. I appreciate the inclusion of the ice optical property case for completeness; however, it's important to note that this issue is not the result of a flawed airborne or satellite measurement, nor is it due to atmospheric or topographic correction. This should be acknowledged in the manuscript.

We will rephrase. The point is that newer ice refractive index measurements, including those from Warren and Brandt (2008), indicate no hook for clean snow, so its presence in an airborne or satellite measurement for clean snow is an error.

Added to l 92 "...,minor and unrelated to the first two"

4. Need to define mu in equation 2

It's defined on L 75, so we will move this definition.

Moved to just below Eq 2

5. Consider adding subsection numbering to section 3 for each case.

We will do this.

Now 3.1 to 3.3

References

Donahue, C. P., Menounos, B., Viner, N., Skiles, S. M., Beffort, S., Denouden, T., ... & Heathfield, D. (2023). Bridging the gap between airborne and spaceborne imaging spectroscopy for mountain glacier surface property retrievals. *Remote Sensing of Environment*, *299*, 113849.

References

Warren, S. G. and Brandt, R. E.: Optical constants of ice from the ultraviolet to the microwave: A revised compilation, Journal of Geophysical Research: Atmospheres, 113, 10.1029/2007JD009744, 2008.

---

## Author Response (AR3)

Dear Dr. Derksen:

We appreciate the attention to this manuscript. Our replies are in red. Your text is in blue and Reviewer 1's comments are in black.

Dear Dr. Bair – I don't find that you have seriously considered the comments of Reviewer 1, as I requested in my previous editorial decision. The Reviewer made reasonable requests to revisit and clarify the description of results in Figure 1c,

We took Reviewer 1's comments seriously. We've copied the relevant text from Reviewer 1 below.

Still the comparison showed in Fig. 1c is a bit strange to me. I would suggest the authors to provide further discussion on this particular comparison. The drop in reflectance for wl<700n is marked here. Given that the surface was flat, I assume that in this case the hooking was due to atmospheric correction or related to the refractive index of ice.

In our initial response to Reviewer 1, we provided an extensive description of the spectroscopic methods used in the field and on the PRISMA data for the reviewer and summarized this in manuscript "The measured and modeled spectra in Figure1a-c and Figures 2a-c are from level and fully snow-covered areas , i.e., no vegetation within the pixel or in adjacent pixels and an optically thick snowpack."

Thus, we agree that the hooking in Fig 1c is due to an atmospheric correction issue, which is discussed in Section 3.1 *Hook caused by atmospheric correction algorithm*. The hook is too large in magnitude to be due to differences in the refractive index of ice.

To emphasize this on l 34, we've added "…likely due to an atmospheric correction error (Section **Error! Reference source not found.**),…"

and add citations for evidence of the blue hook over glacier ice (such as low relief ablation areas), both of which you have dismissed. I took the liberty of revisiting the reviews of the Bohn et al (2024) manuscript, and I have provided the relevant citations below.

Di Mauro, B., Baccolo, G., Garzonio, R., Giardino, C., Massabò, D., Piazzalunga, A., Rossini, M., and Colombo, R.: Impact of impurities and cryoconite on the optical properties of the Morteratsch Glacier (Swiss Alps), The Cryosphere, 11, 2393–2409, https://doi.org/10.5194/tc-11-2393-2017, 2017.

Kokhanovsky A, Di Mauro B and Colombo R (2022) Snow surface properties derived from PRISMA satellite data over the Nansen Ice Shelf (East Antarctica). Front. Environ. Sci. 10:904585. doi: 10.3389/fenvs.2022.904585

Naegeli K., A. Damm, M. Huss, M. Schaepman, and M. Hoelzle, "Imaging spectroscopy to assess the composition of ice surface materials and their impact on glacier mass balance," Remote Sens. Environ., vol. 168, pp. 388–402, Oct. 2015, doi: 10.1016/j.rse.2015.07.006.

Please revise the manuscript accordingly and publication in The Cryosphere can proceed. Best regards,

Chris Derksen

Thank you for providing the references from the anonymous review in Bohn et al. (2024). We've copied the relevant text from Reviewer 1 below and have carefully studied each of the references.

I would also suggest to broaden their discussion also on glacier ice. In my review of Bohn et al. I detailed this issue. The hooking can be observed on glacier ice as well. I think this should be mentioned in this manuscript.

Kokhanovsky et al. (2022) clearly show the hooking problem, but not over exposed glacier ice. The high visible reflectance of the spectra clearly shows optically thick snow, not exposed glacier ice.

Di Mauro et al. (2017) show the hooking issue over ice glacier ice from Hyperion retrievals, but the authors were not able to find a region of clean exposed ice for comparison between the field spectroscopic measurements, which do not show hooking, and those from Hyperion. From Di Mauro et al. (2017), "Unfortunately, we were unable to identify a pure region with clean ice in the ablation zone to compare with ASD spectra on the Hyperion image". This issue was suggested in our previous author's response, "Further we posit that finding clean and level exposed glacier ice over a pixel size of say 30 – 60 m is unusual."

Likewise, Naegeli et al. (2015) show the hook over glacier ice acquired from an aerial imaging spectrometer, however these "bright ice" shown in Fig A1 are clearly dirty, with the light absorbing particles being visible to the naked eye, which is not surprising given the 31 Aug 2013 flight date.

Thus, Kokhanovsky et al. (2022) show snow, not exposed ice and Di Mauro et al. (2017) and Naegeli et al. (2015) show dirty exposed ice. Therefore none of the references show exposed glacier ice with an erroneous hook.

We concede that erroneous hooking could occur over brighter glacier ice (e.g., clean nevé). Thus we have added the following L 111-112 "We also suggest that this erroneous hooking could occur over brighter exposed clean glacier ice, e.g., clean névé."